# DIFFUSION DATASET CONDENSATION: TRAINING YOUR DIFFUSION MODEL FASTER WITH LESS DATA

## ABSTRACT

Diffusion models have achieved remarkable success in various generative tasks, but training them remains highly resource-intensive, often with millions of images and GPU days of computation required. From a data-centric perspective addressing the limitation, we study diffusion dataset condensation as a new challenging problem setting that aims at constructing a "synthetic" sub-dataset with significantly fewer samples than the original dataset for training high-quality diffusion models significantly faster. To the best of our knowledge, we are the first to formally study the dataset condensation task for diffusion models, while conventional dataset condensation focused on training discriminative models. For this new challenge, we further propose a novel **D**iffusion **D**ataset **C**ondensation ($D^2C$) framework, that consists of two phases: *Select* and *Attach*. The *Select* phase identifies a compact and diverse subset via a diffusion difficulty score and interval sampling, upon which the *Attach* phase enhances conditional signals and information of the selected subset by attaching rich semantic and visual representations. Extensive experiments across dataset sizes, model architectures, and resolutions demonstrate that our $D^2C$ can train diffusion models significantly faster with dramatically fewer data while retaining high visual quality. Notably, for the SiT-XL/2 architecture, our $D^2C$ achieves a $100\times$ acceleration, reaching a FID of 4.3 in just 40k steps using only 0.8% of the training data.

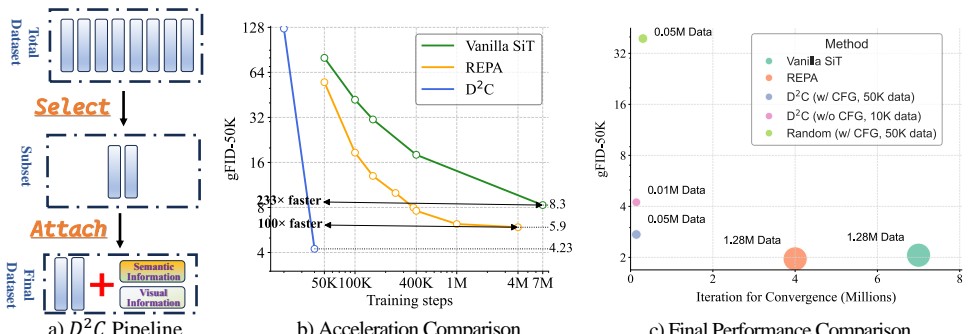

a) $D^2C$ Pipeline  b) Acceleration Comparison  c) Final Performance Comparison

Figure 1: **$D^2C$ framework significantly accelerates diffusion model training with limited data.** **(a)** Overview of our $D^2C$ pipeline, which consists of a *Select* phase that filters a compact and diverse subset via diffusion difficulty score and interval sampling, and an *Attach* phase that enriches samples with semantic and visual information. **(b)** $D^2C$ achieves over $100\times$ faster convergence compared to REPA and over $233\times$ faster than vanilla SiT-XL/2, reaching a FID of 4.3 at just 40k steps. **(c)** Under a strict 4% data budget (0.05M), our method achieves a FID of 2.7 at 180k iterations, demonstrating its strong training efficiency and rapid convergence.

## 1 INTRODUCTION

Generative models, such as score-based (Song et al., 2023b;a; Ho et al., 2020) and flow-based (Liu et al., 2022; Stability.ai, 2024) approaches, have achieved remarkable success in various generative tasks, producing high-quality and diverse data across domains (Karras et al., 2022; Guo et al., 2023).

However, these approaches are notoriously data and compute intensive to train, often requiring millions of samples and hundreds of thousands of iterations to capture complex high-dimensional distributions (Peebles & Xie, 2023; Yu et al., 2025; Ma et al., 2024). The resulting cost presents a significant barrier to broader application and iteration within the AIGC community, making efficient training increasingly important across both academic and industrial settings (Li et al., 2023b; Cui et al., 2022). Recent efforts have improved diffusion training efficiency through various strategies, such as architectural redesigns (Ma et al., 2024; Zheng et al., 2023a; Peebles & Xie, 2023), attention optimization (Bolya & Hoffman, 2023), reweighting strategies (Li et al., 2025; Hang et al., 2023), and representation learning (Yu et al., 2025; Wu et al., 2025). In parallel, data-centric approaches such as patch-based methods (Ding et al., 2024; Wang et al., 2023), Infobatch (Qin et al., 2023) aim to better exploit the potential of existing data. Despite these advances, building a relatively complete "synthetic[1]" subset via dataset condensation (Wang et al., 2018) remains underexplored.

Dataset condensation (Wang et al., 2018; 2022; Yin et al., 2023; Shao et al., 2023) aims to construct a "synthetic" sub-dataset with significantly fewer samples than the original dataset, such that a model trained from scratch on this subset achieves comparable performance to one trained on the full dataset. Unlike data pruning or selection Yang et al. (2023), which passively select existing samples, condensation actively optimizes synthetic data, offering greater potential for aggressive data reduction and training efficiency (Sun et al., 2024). However, all existing methods are designed for discriminative tasks. Compared to discriminative tasks, generative tasks are much more complex and demand higher dataset quality (Manduchi et al., 2024). Applying popular methods (e.g., $SRe^2L$ (Yin et al., 2023)) that have substantiated effective for discriminative tasks to diffusion models presents significant challenges, such as the failure to produce diverse, high-quality outputs with structural and semantic fidelity, leading to degraded results and unstable convergence (see Sec. 4).

We raise a key question: *"Can we train diffusion models dramatically faster with significantly less data, while retaining high generation quality?"* The answer is affirmative. Answering this question holds significant relevance for the further development of visually generative intelligence and is therefore extremely worthwhile to explore. In this paper, we mainly made three contributions.

***First,*** to the best of our knowledge, we are the first to formally study the dataset condensation task for diffusion models, a new challenging problem setting that aims at constructing a "synthetic" sub-dataset with significantly fewer samples than the original dataset for training high-quality diffusion models significantly faster. We address a fundamental academic gap concerning the application of dataset condensation in diffusion models. More specifically, our explorations with the diffusion model provide the first insights into the challenges and potential solutions for applying dataset condensation to vision generation tasks. We note that while conventional dataset condensation made great progress and sometimes use diffusion models to construct a subset, this line of research only focused on training discriminative models instead of generative models.

***Second,*** we propose $D^2C$, a novel two-stage dataset condensation framework tailored for training diffusion models. Our framework addresses the challenges of dataset condensation for diffusion models by decomposing the problem into two key aspects: the *Select* stage identifies an informative, compact, and learnable subset by ranking samples using the diffusion difficulty score derived from a pre-trained diffusion model; the *Attach* stage enriches each selected sample by adding semantic and visual representations, further enhancing the training efficiency while preserving performance.

***Third,*** extensive experiments demonstrate great empirical success that the proposed $D^2C$ can train diffusion models significantly faster with dramatically fewer data while retaining high visual quality, substantiating the effectiveness and scalability. Specifically, $D^2C$ significantly outperforms random sampling and several popular dataset selection and distillation algorithms (e.g., $SRe^2L$ (Yin et al., 2023) and K-Center) across data compression ratios of 0.8%, 4%, and 8%, at resolutions of $256 \times 256$ and $512 \times 512$, and with both SiT (Ma et al., 2024) and DiT (Peebles & Xie, 2023) architectures. In particular, $D^2C$ achieves a FID of 4.3 in merely 40k training steps (w/o classifier-free guidance (CFG) (Ho & Salimans, 2021)) using SiT-XL/2 (Ma et al., 2024), demonstrating a $\mathbf{100\times}$ acceleration over REPA (Yu et al., 2025) and a $\mathbf{233\times}$ speed-up compared to vanilla SiT. Furthermore, it further improves to a FID of 2.7 using only 50k synthesized images with CFG (refer to Fig. 1 (c)).

---

[1]Throughout this paper, "synthetic" subset refers to an *artificially designed-and-enhanced* subset: real samples are ***selected*** and ***augmented/attached*** with semantic and visual representations.

## 2 PRELIMINARIES AND RELATED WORK

In this section, we briefly review diffusion models as well as dataset condensation.

**Diffusion Models.** We briefly introduce the standard latent-space noise injection formulation (Peebles & Xie, 2023), which defines a forward process that gradually perturbs input data $\mathbf{x}_0 \sim q_0(\mathbf{x})$ with Gaussian noise:

$$q_t(\mathbf{x}_t \mid \mathbf{x}_0) = \mathcal{N}(\mathbf{x}_t; \alpha_t \mathbf{x}_0, \sigma_t^2 \mathbf{I}), \tag{1}$$

where $\alpha_t$, $\sigma_t \in \mathbb{R}^+$ are differentiable functions of $t$ with bounded derivatives. The choice for $\alpha_t$ and $\sigma_t$ is referred to as the noise schedule of a diffusion model. After that, we need to train a neural network $\epsilon_\theta(\cdot, \cdot, \cdot)$ to approximate the reverse denoising process (*i.e.*, predict the added noise $\epsilon$) for sampling (see Appendix B for more details). The training objective is to minimize the mean squared error between the predicted and the ground true noise:

$$\mathcal{L}_{\text{diff}} = \mathbb{E}_{\mathbf{x}_0 \sim q_0(\mathbf{x}), \epsilon \sim \mathcal{N}(0, \mathbf{I}), t \sim \mathcal{U}[0,1]} \left[ \|\epsilon - \epsilon_\theta(\mathbf{x}_t, t, \mathbf{c})\|_2^2 \right], \tag{2}$$

Here, $\mathbf{c}$ is a conditional input, such as class labels or text embeddings. In some cases, the prediction target is replaced with the $v$-prediction, which corresponds to flow matching.

**Data-centric Efficient Training.** Various model-side strategies have been proposed to accelerate diffusion model, including architectural enhancements Bolya & Hoffman (2023); Peebles & Xie (2023); Ma et al. (2024), sampling refinements (Lu et al., 2022b; Zheng et al., 2023b; Lu et al., 2022a), and representation-level techniques that leverage pretrained vision features Wu et al. (2025); Yu et al. (2025); Li et al. (2024). However, relatively little has been explored from a data-centirc perspective. In data-centric model training, given an original dataset $\mathcal{D} = (\hat{\mathbf{X}}, \hat{\mathbf{Y}}) = \{(\hat{\mathbf{x}}_i, \hat{y}_i)\}_{i=1}^{|\mathcal{D}|}$, where each $\hat{y}_i$ is the label corresponding to sample $\hat{\mathbf{x}}_i$, dataset compression aims to reduce the size of training data while preserving model performance. Two primary strategies have been extensively studied in this context: dataset pruning and dataset condensation.

*1) Dataset Pruning.* Dataset pruning selects an information-enrichment subset from the original dataset, i.e., $\mathcal{D}^{\text{core}} \subset \mathcal{D}$ with $|\mathcal{D}^{\text{core}}| \ll |\mathcal{D}|$, and directly minimizes the training loss over the subset:

$$\min_\theta \mathbb{E}_{(\mathbf{x}, y) \sim \mathcal{D}^{\text{core}}} \left[ \ell(\phi_{\theta_{\mathcal{D}^{\text{core}}}}(\mathbf{x}), y) \right], \tag{3}$$

where $\ell(\cdot, \cdot)$ denotes the empirical training loss, and $\phi_{\theta_{\mathcal{D}^{\text{core}}}}$ is the model parameterized by $\theta_{\mathcal{D}^{\text{core}}}$. Classical data pruning methods like random sampling, K-Center (Jones et al., 2020), and Herding (Chen & Welling, 2010) can be used with diffusion models, but they offer minimal performance improvements.

*2) Dataset Condensation.* In contrast, dataset condensation aims to synthesize a small, compact, and diverse synthetic dataset $\mathcal{D}^{\mathcal{S}} = (\mathbf{X}, \mathbf{Y}) = \{(\mathbf{x}_j, y_j)\}_{j=1}^{|\mathcal{D}^{\mathcal{S}}|}$ to replace the original dataset $\mathcal{D}$. The synthetic dataset $\mathcal{D}^{\mathcal{S}}$ is generated by a condensation algorithm $\mathcal{C}$ such that $\mathcal{D}^{\mathcal{S}} \in \mathcal{C}(\mathcal{D})$, with $|\mathcal{D}^{\mathcal{S}}| \ll |\mathcal{D}|$. Each $y_j$ corresponds to the synthetic label for the sample $\mathbf{x}_j$.

The key motivation for dataset condensation is to create $\mathcal{D}^{\mathcal{S}}$ such that models trained on it can achieve performance within an acceptable deviation $\eta$ compared to models trained on $\mathcal{D}$. This can be formally expressed as:

$$\sup \left\{ \left| \ell(\phi_{\theta_\mathcal{D}}(\hat{\mathbf{x}}), \hat{y}) - \ell(\phi_{\theta_\mathcal{D}^{\mathcal{S}}}(\hat{\mathbf{x}}), \hat{y}) \right| \right\}_{(\hat{\mathbf{x}}, \hat{y}) \sim \mathcal{D}} \leq \eta, \tag{4}$$

where $\theta_\mathcal{D}$ is the parameter set of the neural network $\phi$ optimized on $\mathcal{D}$: $\theta_\mathcal{D} = \arg\min_\theta \mathbb{E}_{(\hat{\mathbf{x}}, \hat{y}) \sim \mathcal{D}} \left[ \ell(\phi_\theta(\hat{\mathbf{x}}), \hat{y}) \right]$. A similar definition applies to $\theta_\mathcal{D}^{\mathcal{S}}$, which is optimized on the synthetic dataset $\mathcal{D}^{\mathcal{S}}$. Existing methods, such as RDED, MTT, SRe$^2$L, and G-VBSM (Sun et al., 2024; Yin et al., 2023; Shao et al., 2023; 2024; Cazenavette et al., 2022), are primarily designed for discriminative tasks. When applied to diffusion models, these methods generate synthetic images that deviate from the original data distribution, leading to detrimental effects on model training. Visualization of these synthesized images can be found at Appendix J.

## 3 DIFFUSION DATASET CONDENSATION

To enable data-centric efficient training of diffusion models under limited resources, we propose **D**iffusion **D**ataset **C**ondensation (*D²C*), the first unified framework that systematically condenses

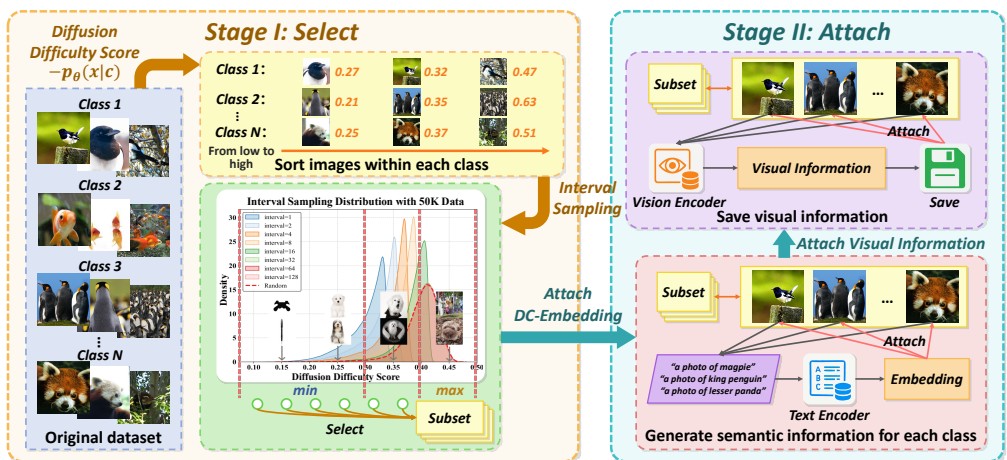

Figure 2: Overview of **D**iffusion **D**ataset **C**ondensation ($D^2C$). $D^2C$ employs a two-stage process: *Select* and *Attach*. The *Select* stage identifies a compact and diverse subset by intervaling sampling using the diffusion difficulty score derived from a pre-trained diffusion model. The *Attach* stage further enriches each selected sample by adding semantic information and visual information.

training data for diffusion models. As illustrated in Fig. 2, this process produces a condensed dataset suitable for efficient diffusion model training. $D^2C$ consists of two stages: *Select* (Sec. 3.1), which identifies a compact set of diverse and learnable real images using *diffusion difficulty score* and *interval sampling* techniques; and *Attach* (Sec. 3.2), which augments each selected image with semantic and visual information to improve generation performance. Finally, we present the novel training paradigm utilizing the condensed dataset generated by $D^2C$ (Sec. 3.3).

## 3.1 *Select*: DIFFICULTY-AWARE SELECTION

In this work, we focus on class-to-image (C2I) synthesis. However, we also show that our framework is applicable to the text-to-image (T2I) setting with only minor changes, as detailed in Appendix G. Given a class-conditioned dataset $\mathcal{D} = \bigcup_{y=1}^{C} \mathcal{D}_y$, where $\mathcal{D}_y = \{x_i\}_{i=1}^{|\mathcal{D}_y|}$ denotes all samples of class $y$, our aim is to select a compact subset for efficient diffusion training. To achieve this, we propose the *diffusion difficulty score* to quantify the denoising difficulty of each sample, followed by our designed *interval sampling* to ensure diversity within the selected subset.

**Diffusion Difficulty Score.** The arrangement of samples from easy to hard is crucial for revealing underlying data patterns and facilitating difficulty-aware selection. Recent work (Li et al., 2023a) demonstrates that diffusion models inherently encode semantic-related class-conditional probability $p_\theta(\mathbf{c}|\mathbf{x})$ through the variational lower bound (i.e., diffusion loss Eq. 2) of $\log p_\theta(\mathbf{x}|\mathbf{c})$ (Ho et al., 2020; Song et al., 2023b). This conditional probability can be formulated as $p_\theta(\mathbf{c}|\mathbf{x}) = \frac{p_\theta(\mathbf{x}|\mathbf{c})p(\mathbf{c})}{\sum_{\hat{\mathbf{c}}} p_\theta(\mathbf{x}|\hat{\mathbf{c}})p(\hat{\mathbf{c}})}$. Intuitively, a larger $p_\theta(\mathbf{c}|\mathbf{x})$ indicates that sample $\mathbf{x}$ can be more confidently identified as belonging to class $\mathbf{c}$, thus suggesting lower learning difficulty. Given the significant computational overhead of the full Bayesian formulation and our focus on estimating sample difficulty, we ignore the denominator part $\sum_{\hat{\mathbf{c}}} p_\theta(\mathbf{x}|\hat{\mathbf{c}})p(\hat{\mathbf{c}})$ of the calculation. Since the category $y \sim U\{0, \cdots, C\}$ ($C$ denotes the class number) is obtained by uniform sampling, and assuming $\sup\{|\mathbb{E}_{\hat{\mathbf{c}}}[p_\theta(\mathbf{x_1}|\hat{\mathbf{c}})] - |\mathbb{E}_{\hat{\mathbf{c}}}[p_\theta(\mathbf{x_2}|\hat{\mathbf{c}})]|\}_{\mathbf{x_1},\mathbf{x_2} \sim \mathcal{D}^{\mathcal{X}}} \leq \eta$ ($\mathcal{D}^{\mathcal{X}}$ denotes the all images in the dataset), we define the diffusion difficulty score based on the class-conditional probability $p_\theta(\mathbf{c}|\mathbf{x}) = \frac{p_\theta(\mathbf{x}|\mathbf{c})}{\sum_{\mathbf{c}} p_\theta(\mathbf{x}|\hat{\mathbf{c}})} \propto p_\theta(\mathbf{x}|\mathbf{c})$ ($\sum_{\mathbf{c}} p_\theta(\mathbf{x}|\hat{\mathbf{c}})$ can be viewed as a constant):

$$s_{\text{diff}}(\mathbf{x}) = -p_\theta(\mathbf{c}|\mathbf{x}) \propto -p_\theta(\mathbf{x}|\mathbf{c}) = -\log\left(\exp\left(-\mathbb{E}_{\epsilon \sim \mathcal{N}(0,\mathbf{I}), t \sim \mathcal{U}[0,1]}\left[\|\epsilon - \epsilon_\theta\left(\alpha_t \mathbf{x} + \sigma_t \epsilon, t, \mathbf{c}\right)\|_2^2\right]\right)\right),$$
(5)

The higher the score $s_{\text{diff}}(\mathbf{x})$, the more difficult it is, and the lower the score $s_{\text{diff}}(\mathbf{x})$, the easier it is. To simplify our presentation, we define the diffusion loss $-p_\theta(\mathbf{x}|\mathbf{c})$ as the diffusion difficulty score.

By computing $s_{\text{diff}}(x)$ for all training samples, we construct a ranked dataset. As shown in Fig. 3, these scores exhibit a skewed unimodal distribution. Selecting the easiest samples (*Min*) yields a subset dominated by clean, background-simple images with high learnability but limited diversity.

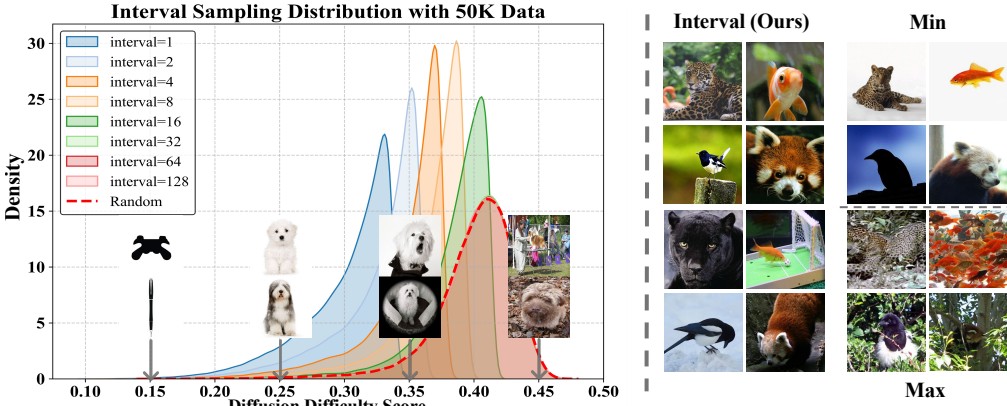

Figure 3: Overview of **D**iffusion **D**ataset **C**ondensation (**$D^2C$**). $D^2C$ employs a two-stage process: *Select* and *Attach*. The *Select* stage identifies a compact and diverse subset by intervaling sampling using the diffusion difficulty score derived from a pre-trained diffusion model. The *Attach* stage further enriches each selected sample by adding semantic information and visual information.

In contrast, selecting only the highest-score samples (*Max*) results in cluttered, noisy, and ambiguous images that are difficult to optimize. Meanwhile, many samples lie in the middle range, offering moderate learnability but richer contextual information. Selecting an appropriate value within this range is therefore critical; we provide a more detailed discussion in Appendix H.1.

**Interval Sampling.** To balance diversity and learnability, we propose an *interval sampling* strategy. Specifically, we sort its images $\mathcal{D}_y$ within each class $y$ in ascending order of $s_{\mathrm{diff}}(x)$ and select samples at a fixed interval $k$: $\mathcal{D}_{\mathrm{IS}} = \bigcup_{y=1}^{C} \left\{ x^{(i)} \in \mathcal{D}_y \mid i \in \{0, k, 2k, \dots\} \right\}$, where $\mathcal{D}_{\mathrm{IS}}$ denotes the selected subset constructed by interval sampling, $k$ is the fixed sampling interval, and $x^{(i)}$ is the $i$-th sample in the sorted list (e.g., $x^{(0)}$ corresponds to the sample with the lowest diffusion difficulty score). Interval sampling with a larger interval $k$ promotes diversity in the sampled data while potentially hindering learnability. As shown in Fig. 3 (Left), this trade-off arises from a shift in the sample distribution: a larger $k$ leads to a reduction in the number of easy samples and a corresponding increase in the representation of standard and difficult samples.

**Extended Discussion.** Training exclusively on the easiest (Min) or the hardest (Max) samples is suboptimal. Instead, a balanced curriculum comprising easy, medium, and difficult examples yields a training subset that is both learnable and diverse, ultimately leading to stronger generative performance. We further offer more discussions and insights on interval sampling in Appendix H.2.

### 3.2 *Attach*: SEMANTIC AND VISUAL INFORMATION ENHANCEMENT

To further enhance the information richness of the selected data, we introduce the *Attach* phase that augments each instance with semantic and visual information. While the *Select* phase generates a compact and informative subset, the achievable performance ceiling based solely on this phase remains limited. Consequently, we inject more comprehensive *semantic* and *visual* representations into the selected subset to further bolster our method's generalization capability.

**Dual Conditional Embedding (DC-Embedding).** Existing C2I synthesis methods (Peebles & Xie, 2023; Ma et al., 2024) commonly rely on class embeddings trained from scratch, which often fail to effectively capture inherent semantic information (see Appendix I.1). We enrich the class embedding by incorporating text representations derived from a pre-trained text encoder (e.g., T5-encoder (Ni et al., 2021)). For each class $c \in \{1, \dots, C\}$, a descriptive prompt $P(c)$ (e.g., *"a photo of a cat"*) is encoded by a pre-trained text encoder $f_{\mathrm{text}}$, yielding its corresponding text embedding $t_c$ and text mask $t_{\mathrm{mask}}$:

$$t_c, t_{\mathrm{mask}} = f_{\mathrm{text}}(P(c)), \tag{6}$$

The resulting text embedding and text mask are stored on disk as attached text information alongside the subset $\mathcal{D}_{\mathrm{IS}}$ generated in the preceding phase, ready for import during formal training. During the formal training, as illustrated in Fig. 5, the text embedding $t_c$ and the text mask $t_{\mathrm{mask}}$ undergo a 1D

**Training Iteration**

Figure 4: **$D^2C$ improves visual quality under tight data budgets.** We compare Random sampling and $D^2C$ on DiT-L/2 at 10k and 50k data budgets, and neither setting uses classifier-free guidance.

convolution and are fused with a learnable class embedding $e_c$ using a residual MLP:

$$\tilde{t}_c = \text{Conv1d}(t_c \times t_{\text{mask}}), \quad y_{\text{text}} = \text{MLP}(\tilde{t}_c) + \tilde{t}_c + e_c. \tag{7}$$

This resulting vector $y_{\text{text}}$ then serves as a semantic conditioning token for the conditional diffusion model. Compared to using simple class embeddings alone, this formulation offers richer semantic information while retaining the learnability of class embeddings.

**Visual Information Injection.** While semantic information aids in distinguishing inter-class structure, it often fails to capture the intra-class variability essential for high-fidelity generation. To address this, we integrate instance-specific visual representations into the attached information. For each image $x \in \mathbb{R}^{3 \times H \times W}$, a pre-trained vision encoder $f_{\text{vis}}$ (e.g., DINOv2 (Oquab et al., 2023)) extracts patch-level semantic representations:

$$y_{\text{vis}} = f_{\text{vis}}(x) \in \mathbb{R}^{N \times d_{\text{text}}} \tag{8}$$

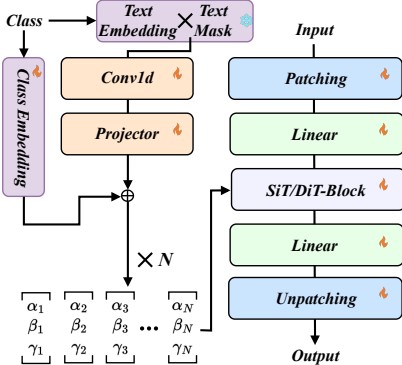

Figure 5: Overview of DC-Embedding.

where $N$ is the number of image patches and $d_{\text{text}}$ is the feature dimension. We retain the first $h$ (i.e., number of tokens in the diffusion transformer) tokens of $y_{\text{vis}}$ to form a compact representation of the dominant structure: $y_{\text{vis}} = y_{\text{vis}}[:h,:] \in \mathbb{R}^{h \times d_{\text{text}}}$. As outlined in REPA (Yu et al., 2025), this visual information provides a semantic prior for the diffusion model and thus significantly benefits data-centric efficient training. Similar to the text information $y_{\text{text}}$, the visual information $y_{\text{vis}}$ is also stored on disk as attached metadata alongside the selected subset $\mathcal{D}_{\text{IS}}$.

### 3.3 $D^2C$ TRAINING PROCESS

Here, we detail the training process of the diffusion model using our condensed dataset, which comprises a compact subset selected during the *Select* phase and subsequently enriched with semantic and visual information during the *Attach* phase. Our goal is to fully leverage the information existed within our condensed dataset to accelerate training without compromising performance.

We employ a conditional diffusion model $\mathcal{D}_\theta$ and, as an example, utilize the optimization objective of score-based diffusion models: predicting the added noise $\epsilon$ from the perturbed latent input $\mathbf{x}_t$ at time step $t$, conditioned on the text information $y_{\text{text}}$ and the class label $y$. The new denoising loss is defined as $\mathcal{L}_{\text{diff}}^{\text{new}} = \mathbb{E}_{\mathbf{x}_0 \sim q_0(\mathbf{x}), \epsilon \sim \mathcal{N}(0,\mathbf{I}), t \sim \mathcal{U}[0,1]} \left[ \|\epsilon - \epsilon_\theta(\mathbf{x}_t, t, y, y_{\text{text}})\|_2^2 \right]$, where the specific injected forms of $y$ and $y_{\text{text}}$ can be found in Sec. 3.2. Then, to maximize the utilization of visual information, we adopt the same formulation as REPA (Yu et al., 2025), which involves aligning the encoder's output (i.e., the decoder's input) within the diffusion model with the visual representation $y_{\text{vis}} = \{v_i\}_{i=1}^h$. Concretely, from a designated intermediate layer of the diffusion backbone, we obtain token features $\{h_i \in \mathbb{R}^d\}_{i=1}^h$. A projection head $\phi$ maps these tokens from $\mathbb{R}^d$ to $\mathbb{R}^{d_{\text{text}}}$, and we compute a semantic alignment loss:

$$\mathcal{L}_{\text{proj}} = -\frac{1}{h} \sum_{i=1}^{h} \left\langle \frac{\phi(h_i)}{\|\phi(h_i)\|}, \frac{v_i}{\|v_i\|} \right\rangle. \tag{9}$$

Table 1: Comparison of gFID-50K across various dataset condensation methods and data budgets using DiT-L/2 and SiT-L/2 on ImageNet 256×256. We use CFG=1.5 for evaluation. $D^2C$ surpasses other methods at all settings.

| Data Budget | Iter. | DiT-L/2 | | | | SiT-L/2 | | | |
|---|---|---|---|---|---|---|---|---|---|
| | | Random | K-Center | Herding | $D^2C$ | Random | K-Center | Herding | $D^2C$ |
| 0.8% (10K) | 100k | 35.86 | 50.77 | 40.75 | **4.20** | 4.35 | 14.77 | 22.96 | **3.98** |
| 0.8% (10K) | 300k | 4.19 | 13.5 | 22.35 | **4.13** | 4.33 | 13.58 | 22.55 | **3.98** |
| 4.0% (50K) | 100k | 36.78 | 69.86 | 32.38 | **14.81** | 31.13 | 61.66 | 29.11 | **11.21** |
| 4.0% (50K) | 300k | 11.55 | 38.54 | 22.44 | **5.99** | 14.18 | 39.69 | 22.44 | **5.66** |
| 8.0% (100K) | 100k | 41.02 | 71.31 | 36.37 | **22.55** | 36.64 | 66.96 | 32.3 | **15.01** |
| 8.0% (100K) | 300k | 11.49 | 37.35 | 15.23 | **6.49** | 12.56 | 39.08 | 16.17 | **5.65** |

Table 2: Comparison with a strict data budget 0.8% (10K) on ImageNet 512×512. We use CFG=1.5 for evaluation. $D^2C$ surpasses random sampling at all settings.

| Model | Method | Iter. | gFID↓ | sFID↓ | Inception Score↑ | Precision↑ |
|---|---|---|---|---|---|---|
| DiT-L/2 | Random | 100k | 24.8 | 11.9 | 74.3 | 0.65 |
| DiT-L/2 | $D^2C$ (Ours) | 100k | **14.8** | **6.9** | **109.2** | **0.63** |
| DiT-L/2 | Random | 300k | 17.1 | **12.8** | 130.6 | 0.64 |
| DiT-L/2 | $D^2C$ (Ours) | 300k | **5.8** | 15.1 | **318.9** | **0.77** |
| SiT-L/2 | Random | 100k | 13.3 | 22.8 | 197.1 | 0.69 |
| SiT-L/2 | $D^2C$ (Ours) | 100k | **9.1** | **14.3** | **261.7** | **0.72** |
| SiT-L/2 | Random | 300k | 5.0 | 13.6 | **316.9** | 0.76 |
| SiT-L/2 | $D^2C$ (Ours) | 300k | **4.22** | **11.6** | 289.7 | **0.79** |

This loss encourages the model to align its encoder's output with visual representations, promoting localized realism and spatial consistency (Oquab et al., 2023) in generation.

**Overall Training Objective.** The final training loss combines the denoising objective and the semantic alignment term (with the balance weight $\lambda$ is set to 0.5 by default):

$$\mathcal{L}_{\text{total}} = \mathcal{L}_{\text{diff}} + \lambda \mathbb{E}_{\mathbf{x}, \epsilon \sim \mathcal{N}(0, \mathbf{I}), t \sim \mathcal{U}[0,1], y, y_{\text{text}}, y_{\text{vis}}} \left[ \mathcal{L}_{\text{proj}} \right]. \tag{10}$$

This unified training strategy enables $D^2C$ to effectively learn from limited yet enhanced data, offering a practical solution for efficient diffusion training under resource-constrained settings.

## 4 EXPERIMENTS

In this section, we validate the performance of $D^2C$ and analyze the contributions of its components through extensive experiments. In particular, we aim to answer the following questions:

*1)* Can $D^2C$ improve training speed and reduce data usage of diffusion models?

*2)* Does $D^2C$ generalize well across backbones, data scales, and resolutions?

*3)* How do $D^2C$'s components and hyperparameter choices affect its overall effectiveness?

### 4.1 SETUP

**Experiment settings.** We conduct experiments on the ImageNet-1K dataset (Russakovsky et al., 2015), using subsets of 10K, 50K, and 100K images, corresponding to 0.8%, 4%, and 8% of the full dataset, respectively. All images are center-cropped and resized to 256×256 and 512×512 resolutions using the ADM Dhariwal & Nichol (2021) preprocessing pipeline. Furthermore, we use [·]-L/2 and [·]-XL/2 architectures in both DiT Peebles & Xie (2023) and SiT Ma et al. (2024) backbones, following the standard settings outlined in Ma et al. (2024). More details on implementation and training can be found in Appendix C.

**Evaluation and baselines.** We train models from scratch on the collected subset and evaluate them using gFID (Heusel et al., 2017), sFID, Inception Score (Salimans et al., 2016) and Precision, adhering to standard evaluation protocols (Dhariwal & Nichol, 2021; Peebles & Xie, 2023; Ma et al.,

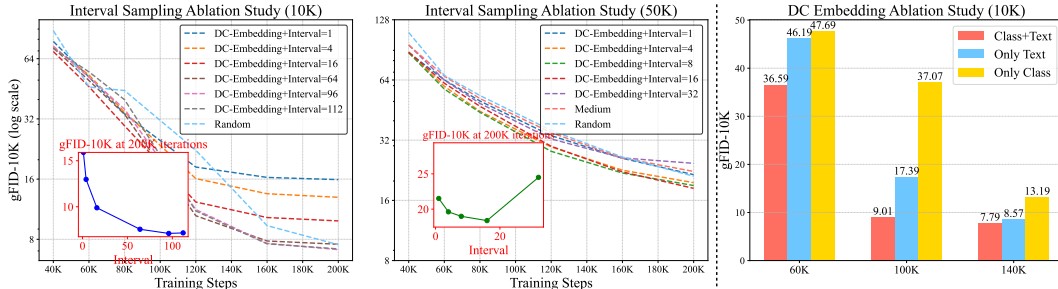

Figure 6: **Left: Interval-sampling ablation.** Small $k$ speeds early training. The best final gFID-10K appears at $k=96$ for the 10K budget and $k=16$ for the 50K budget, roughly scaling with data size. **Right: DC-Embedding ablation at 10K.** Combining text and class embeddings outperforms either alone; "Only Class" denotes the baseline that injects class embeddings only.

2024). We compare our method against REPA Yu et al. (2025), and various data condensation and selection baselines, including SRe$^2$L Yin et al. (2023), Herding, K-Center, and random sampling, using SiT and DiT architectures (Ma et al., 2024; Peebles & Xie, 2023). Further details regarding evaluation metrics and baseline methods can be found in Appendix D and E.

## 4.2 Main Result

**Training Performance and Speed.** We evaluate $D^2C$ on SiT-XL/2 using 10K and 50K data budgets, comparing its performance against REPA and a vanilla SiT model trained on the full ImageNet dataset (a 1.28M data budget), as well as random selection with 10K and 50K data budgets. As shown in Table 3 and Fig. 1 (b), our method achieves a gFID-50K of 4.23 at only 40K iterations with 10K training data. In contrast, REPA requires 4 million steps and the vanilla SiT model needs 7 million steps to reach comparable performance, representing an acceleration of over **100×** and **233×**, respectively. Under a 4% data budget (50K) with CFG set to 1.5, our method achieves an FID of 2.78 at 180K steps, further demonstrating significant data and compute efficiency (Fig. 1 (c)). Moreover, Fig. 4 presents a visual comparison between random selection and our $D^2C$ at 10K and 50K data sizes. Our method demonstrates superior visual quality compared to the baseline and generates higher-quality images, even during the early iterations of training.

**Comparison on ImageNet 256×256.** We compare $D^2C$ with random sampling, Herding (Chen et al., 2023), K-Center (Jones et al., 2020), and SRe$^2$L (Yin et al., 2023) under various data budgets and backbones. As shown in Table 1, $D^2C$ consistently achieves the lowest FID across all settings. For instance, using only 0.8% of the data and 100K iterations with early stopping, our method achieves a gFID-50K of 4.20 on DiT-L/2 and 3.98 on SiT. These results demonstrate the superiority of our approach over existing methods. Notably, SRe$^2$L, which performs well in classification task, fails on this generative task (see Table 4) due to its focus on category-discriminative features. Similarly, geometry-based methods like Herding and K-Center, along with random sampling, prove inadequate for achieving efficient and high-performing training.

Table 3: Comparison of acceleration algorithms on ImageNet-1K.

| Model | Training Set | Iter. | gFID↓ |
|---|---|---|---|
| DiT L-2 | 1.28M | 400k | 23.3 |
| + REPA | 1.28M | 400k | 15.6 |
| + $D^2C$ | **0.05M** | **10k** | **14.81** |
| + $D^2C$ | **0.01M** | **10k** | **4.2** |
| SiT L-2 | 1.28M | 400k | 18.8 |
| + REPA | 1.28M | 700k | 8.4 |
| + $D^2C$ | **0.01M** | **80k** | **7.07** |
| SiT XL-2 | 1.28M | 7M | 8.3 |
| + REPA | 1.28M | 4M | 5.9 |
| + $D^2C$ | **0.01M** | **40k** | **4.3** |
| + $D^2C$ | **0.05M** | **180k** | **2.78** |

**Comparison on ImageNet 512×512.** As shown in Table 2, $D^2C$ achieves a gFID of 5.8 on DiT-L/2, a significant improvement over the 17.1 achieved by random sampling at 300k iterations under the ImageNet 512×512 settings. On SiT-L/2, similar improvements are observed. These results demonstrate that $D^2C$ generalizes well to higher resolutions.

## 4.3 Ablation Study

**Ablation on *Select* Phase.** We investigate the impact of the interval value $k$ in the *Select* phase, as shown in Fig. 6 (Left). Using a small value accelerates early training by prioritizing min-loss sam-

Table 4: $D^2C$ vs. SRe$^2$L (Yin et al., 2023) on ImageNet 256×256 with a data budget 0.8% (10K).

| Model | Method | gFID↓ | sFID↓ | Inception Score↑ | Precision↑ |
|---|---|---|---|---|---|
| DiT-L/2 | SRe$^2$L | 104.2 | 20.2 | 14.1 | 0.20 |
| DiT-L/2 | $D^2C$ (Ours) | **4.2** | **11.0** | **283.6** | **0.72** |
| SiT-L/2 | SRe$^2$L | 82.3 | 19.8 | 18.1 | 0.27 |
| SiT-L/2 | $D^2C$ (Ours) | **3.9** | **10.7** | **289.7** | **0.73** |

Table 5(a) Ablation studies on the Select and Attach phases. Sel.: Select. Vis.: Vision.

| Model | Sel. | DC Emb. | Vis. Emb. | gFID↓ |
|---|---|---|---|---|
| DiT-L/2 | ✗ | ✗ | ✗ | 37.07 |
| DiT-L/2 | ✗ | ✓ | ✓ | 8.79 |
| DiT-L/2 | ✓ | ✗ | ✗ | 14.96 |
| DiT-L/2 | ✓ | ✗ | ✓ | 10.37 |
| DiT-L/2 | ✓ | ✓ | ✗ | 9.01 |
| DiT-L/2 | ✓ | ✓ | ✓ | **7.62** |

Table 5(b) A breakdown of the computational overhead for sub-processes in D$^2$C. Compared to the REPA baseline, the additional scoring time is negligible, demonstrating D$^2$C's efficiency.

| Method | Score Model | Score Time | Iter. | Train Time | gFID↓ |
|---|---|---|---|---|---|
| REPA | N/A | N/A | 4M | 750h | 5.9 |
| D$^2$C (w/o select) | N/A | N/A | **0.04M** | 7.4h | 5.6 |
| D$^2$C (w/ select) | From Scratch | **1.9h** | **0.04M** | (7.4+26.2)h | 4.9 |
| D$^2$C (w/ select) | Pretrained | 2.1h | **0.04M** | 7.4h | **4.3** |

ples, which are simpler and easier to learn. However, the limited diversity of such samples leads to degraded performance in later stages, eventually being overtaken by settings with moderate interval values. In contrast, large intervals or random selection introduce excessive max-loss or uncurated samples, destabilizing training (Fig. 3). As $k$ increases, we observe that gFID-10K first decreases and then worsens, revealing an optimal trade-off between diversity and learnability. Empirically, the best results are achieved with an interval of 96 for the 10K budget and 16 for 50K, approximately following the ratio of data budgets (50K/10K). Table 5a further shows that using the *Select* stage alone reduces gFID from 37.07 to 14.96, underscoring its effectiveness and usefulness.

**Ablation on *Attach* Phase.** We evaluate *Attach* from two angles. First, as shown in Fig. 6 (Right), DC embedding consistently outperforms using either alone under a 10K budget, with text-only better than class-only, indicating richer semantics from textual descriptions. Second, Table 5a shows steady gains from the injection modules: baseline gFID-10K is 14.96, adding only visual information reaches 10.37, adding only DC embedding reaches 9.01, and combining both achieves the best 7.62. Appendix I.2 further ablates the visual encoder and demonstrates the robustness of our approach.

**Effect of pretrained diffusion models and wall-clock cost.** Our D$^2$C pipeline does not inherently require a powerful pretrained model. As shown in Table 5b, when the scoring network is a strong DiT-XL/2 with base gFID 2.27 from (Peebles & Xie, 2023), D$^2$C reaches an FID of 4.3; with a weaker DiT-L/2 that we trained from scratch achieving a base gFID of 11.5, it reaches 4.9. Using only the *Attach* stage, without *Select*, still reaches 5.6 and surpasses REPA at 5.9. In wall-clock terms, the *Attach*-only variant finishes in 7.4h, which is 0.99% of REPA's 750h and about 101× faster. With a pretrained scorer, the end-to-end pipeline totals 9.5h, with 2.1h for scoring and 7.4h for training; this is 1.27% of REPA and about 79× faster. With a scorer trained from scratch, the pipeline totals 35.5h, with 1.9h for scoring, 26.2h for training the scorer, and 7.4h for diffusion training; this is 4.7% of REPA and about 21× faster. These results show that whether the scorer is strong, weak, or omitted, D$^2$C consistently accelerates diffusion training while maintaining competitive quality.

## 5 CONCLUSION

In this paper, we introduce $D^2C$, the first dataset condensation framework that significantly accelerates diffusion model training for generative tasks. Our pipeline comprises two key phases, *Select* and *Attach*. In the *Select* phase, we leverage a diffusion difficulty score and interval sampling to obtain a subset that is both compact and diverse. In the *Attach* phase, we augment this subset with critical semantic and visual information, yielding large training speedups and robust performance. As a pioneering approach in this direction, $D^2C$ achieves 100–233× faster training than strong baselines. We believe this work will inspire and motivate further research in this promising area.

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

## A    LLM Usage Statement

The manuscript text was polished with a large language model. The authors curated and verified all factual content such as methods, key metrics, results and technical specifications. The model served only as a writing aid to improve clarity, coherence and fluency, and it did not design methods, run experiments, analyze results or draw conclusions. All content was reviewed and approved by the authors to ensure accuracy.

## B    Additional Descriptions of Diffusion Models

This section reviews the fundamentals of the Denoising Diffusion Probabilistic Model (DDPM) (Ho et al., 2020). The DDPM framework consists of a fixed forward process that incrementally perturbs the input data with noise, and a learned reverse process trained to iteratively denoise the data, thereby learning the target distribution. Specific architectural details of our implementation are summarized in Appendix B.2.

### B.1    Denoising Diffusion Probabilistic Model (DDPM)

The DDPM framework models data generation via a discrete-time Markov chain that progressively adds Gaussian noise to a data sample $x_0 \sim p(x)$. The forward process is defined as:

$$q(x_t \mid x_{t-1}) = \mathcal{N}(x_t; \sqrt{1 - \beta_t} x_{t-1}, \beta_t \mathbf{I}), \tag{11}$$

where $\beta_t \in (0, 1)$ are predefined variance schedule parameters controlling the noise level at each time step $t \in [1, 2, ..., T]$, and $\mathbf{I}$ is the identity matrix.

For simplicity, we define $\alpha_t = 1 - \beta_t$, and denote the cumulative product $\bar{\alpha}_t = \prod_{i=1}^{t} \alpha_i$. The reverse process, which is learned by the model $\theta$, can be defined as:

$$p_\theta(x_{t-1} \mid x_t) = \mathcal{N}\left(x_{t-1}; \frac{1}{\sqrt{\alpha_t}}\left(x_t - \frac{\beta_t}{\sqrt{1 - \bar{\alpha}_t}} \epsilon_\theta(x_t, t)\right), \Sigma_\theta(x_t, t)\right), \tag{12}$$

where $\epsilon_\theta(x_t, t)$ denotes the predicted noise from a neural network. The covariance $\Sigma_\theta(x_t, t)$ is typically set to $\sigma_t^2 \mathbf{I}$, where $\sigma_t^2$ can be either fixed ($\sigma_t^2 = \beta_t$) or learned through interpolation $\sigma_t^2 = (1 - \bar{\alpha}_{t-1})/(1 - \bar{\alpha}_t)\beta$.

A simplified training objective minimizes the prediction error between true and estimated noise:

$$\mathcal{L}_{\text{simple}} = \mathbb{E}_{x_0, \epsilon, t}\left[\|\epsilon - \epsilon_\theta\left(\sqrt{\bar{\alpha}_t} x_0 + \sqrt{1 - \bar{\alpha}_t}\epsilon, t\right)\|^2\right]. \tag{13}$$

In addition to the simple objective, improved variants include learning the reverse variance $\Sigma_\theta(x_t, t)$ jointly with the mean, which leads to a variational bound loss of the form:

$$\mathcal{L}_{\text{vlb}} = \exp\left(v \log \beta_t + (1 - v) \log \tilde{\beta}_t\right). \tag{14}$$

Here, $v$ is an element-wise weight across model output dimensions. When $T$ is sufficiently large and the noise schedule is carefully chosen, the terminal distribution $p(x_T)$ approximates an isotropic Gaussian. Sampling is then performed by iteratively applying the learned reverse process to recover the data sample from pure noise.

### B.2    Diffusion Transformer Architecture

Our model implementation closely follows the design of DiT (Peebles & Xie, 2023) and SiT (Ma et al., 2024), which extend the vision transformer (ViT) architecture (Dosovitskiy et al., 2020) to generative modeling. An input image is first split into patches, reshaped into a 1D sequence of length $N$, and then processed through transformer layers. To reduce spatial resolution and computational cost, we follow prior work (Peebles & Xie, 2023; Ma et al., 2024) and encode the image into a latent tensor $z = E(x)$ using a pretrained encoder $E$ from the stable diffusion VAE (Kingma, 2013).

In contrast to the standard ViT, our transformer blocks include time-aware adaptive normalization layers known as AdaIN-zero. These layers scale and shift the hidden state in each attention block according to the diffusion timestep and conditioning signals. During training, we also add an auxiliary multilayer perceptron (MLP) head that maps the hidden state to a semantic target representation space, such as DINOv2 (Oquab et al., 2023) or CLIP features (Radford et al., 2021). This head is used only for training-time supervision in our alignment loss and does not affect sampling or inference.

## C    Hyperparameters and Implementation Details

**Select Phase Settings.** In the *Select* phase, we adopt a pre-trained DiT-XL/2 model (Peebles & Xie, 2023) as the scoring network and use the diffusion loss (*w.r.t.*, mean squared error) as the scoring metric. To construct subsets of different sizes, we apply interval sampling with $k = 96$ for the 10K subset, $k = 16$ for the 50K subset, and $k = 10$ for the 100K subset. Each subset is constructed in a class-wise manner, selecting 10, 50, and 100 samples per class respectively.

**Attach Phase Settings.** In the *Attach* phase, we implement dual conditional embeddings. For textual conditioning, we use a T5 encoder (Ni et al., 2021) with captions truncated to 16 tokens, producing embeddings of dimension 2048. For visual conditioning, we adopt DINOv2-B (Oquab et al., 2023) as the visual encoder. The number of visual tokens $h$ is set to 256, and each token has a feature dimension of 768.

**Training Settings.** In the *Training* phase, we use the Adam optimizer with a fixed learning rate of 1e-4 and $(\beta_1, \beta_2) = (0.9, 0.999)$, without applying weight decay. We employ mixed-precision (fp16) training with gradient clipping. Latent representations are pre-computed using the stable diffusion VAE (Kingma, 2013), and decoded via its native decoder. All experiments are conducted on either 8 NVIDIA A800 80GB GPUs or 8 NVIDIA RTX 4090 24GB GPUs. We use a batch size of 256 with a $256 \times 256$ resolution in Fig. 1, and a $512 \times 512$ resolution in Table 2. All other experiments use a batch size of 128 and a default image resolution of $256 \times 256$.

## D    Evaluation Details

We adopt several widely used metrics to evaluate generation quality and diversity:

- **gFID** (Heusel et al., 2017) computes the Fréchet distance between the feature distributions of real and generated images. Features are extracted using the Inception-v3 network (Szegedy et al., 2016).

- **sFID** (Nash et al., 2021) extends FID by leveraging intermediate spatial features from the Inception-v3 model to better capture spatial structure and style in generated images.

- **IS** (Salimans et al., 2016) evaluates both the quality and diversity of generated samples by computing the KL-divergence between the conditional label distribution and the marginal distribution over predicted classes, using softmax-normalized logits.

- **Precision and Recall** (Kynkäänniemi et al., 2019) respectively measure sample realism and diversity, quantifying how well generated samples cover the data manifold and vice versa.

## E    Baseline Setting

We evaluate our method against two categories of baselines:

**Diffusion models trained on selected or condensed subsets.** These include SiT and DiT backbones trained from scratch on 10K, 50K, and 100K subsets obtained via the following strategies:

- **Random Sampling.** A naive baseline that randomly selects a fixed number of real samples without any guidance.

- **Herding** (Chen & Welling, 2010). A geometry-based method that selects samples to approximate the global feature mean, ensuring representative coverage.

- **K-Center** (Jones et al., 2020). A diversity-focused algorithm that iteratively selects samples maximizing the minimum distance from the selected set, promoting broad coverage of the feature space.
- **SRe²L** (Yin et al., 2023). A dataset condensation method that synthesizes class-conditional data through a multi-stage pipeline. Originally proposed for classification tasks, we adapt it to the diffusion setting by applying class-wise condensation to real images and training a diffusion model on the resulting synthetic subset. Visualizations of the synthesized samples and corresponding training results are provided in Appendix J.

**Diffusion models trained on the full dataset.** These baselines are trained with access to the entire training set, without data reduction:

- **SiT** (Ma et al., 2024). A transformer-based diffusion model that reformulates denoising as continuous stochastic interpolation, enabling faster training and improved efficiency under full-data settings.
- **REPA** (Yu et al., 2025). A model-side regularization method that aligns intermediate features of diffusion transformers with patch-wise representations from strong pretrained visual encoders (e.g., DINOv2-B (Oquab et al., 2023), MAE (He et al., 2022), MoCov3 (He et al., 2020)) using a contrastive loss. It retains the full dataset and improves convergence and generation quality via early-layer representation guidance.

## F  FRAMEWORK DESIGN AND IMPLEMENTATION

We introduce $D^2C$, a framework for constructing compact yet effective training subsets for diffusion models under stringent data budgets. Our approach is motivated by two complementary intuitions: (1) that the contribution of training samples is non-uniform, as some are more informative than others; and (2) that generative training benefits from semantically enriched conditioning. These insights directly inform the two core stages of our framework. First, a *Select* stage ranks training examples by a difficulty score computed via a pretrained class-conditional diffusion model. Second, an *Attach* stage enriches the selected data by injecting textual and visual priors. The complete pipeline is summarized in Algorithm 1.

---

**Algorithm 1** $D^2C$: Diffusion Dataset Condensation

---

**Require:** Full dataset $\mathcal{D} = \{(x_i, c_i)\}_{i=1}^N$, interval $k$, text encoder $f_{\text{text}}$, visual encoder $f_{\text{vis}}$
    // *Each $x_i$ is an image, and $c_i \in \{1, \ldots, C\}$ is the class label.*
 1: **// Phase 1: Select**
 2: Compute difficulty score $s_{\text{diff}}$ for all $(x_i, c_i) \in \mathcal{D}$
 3: For each class $c$, sort $\mathcal{D}_c = \{x_i \mid c_i = c\}$ by $s_{\text{diff}}$ descending
 4: Select every $k$-th sample (Interval Sampling) in sorted $\mathcal{D}_c$ to form $\mathcal{D}_{\text{select}}$
 5: **// Phase 2: Attach**
 6: **for** each $(x, c) \in \mathcal{D}_{\text{select}}$ **do**
 7:     Generate class prompt $P(c)$ (e.g., "a photo of a `label`")
 8:     Extract text embedding: $(t_c, t_{\text{mask}}) \leftarrow f_{\text{text}}(P(c))$
 9:     Extract visual feature: $y_{\text{vis}} \leftarrow f_{\text{vis}}(x)$
10:     Store triplet $(x, c, t_c, t_{\text{mask}}, y_{\text{vis}})$ into $\widetilde{\mathcal{D}}$
11: **end for**
12: **Return** enriched dataset $\widetilde{\mathcal{D}}$ for diffusion model training

---

## G  EXPLORATION ON TEXT-TO-IMAGE GENERATION

We further examine the applicability of the $D^2C$ framework to text-to-image generation. The *Select* phase requires only a minimal change: replace the class condition in Eq. 5 with a text condition, i.e., $s_{\text{diff}}^{text}(x) = -p_\theta(x \mid \text{text})$. Using SDXL to score LAION text–image pairs, we observe a difficulty distribution similar to the class-conditional case (Fig. 7; see also Fig. 3 and Fig. 8 (right)). Low-score samples tend to exhibit simple structures, high-score samples often contain complex or cluttered

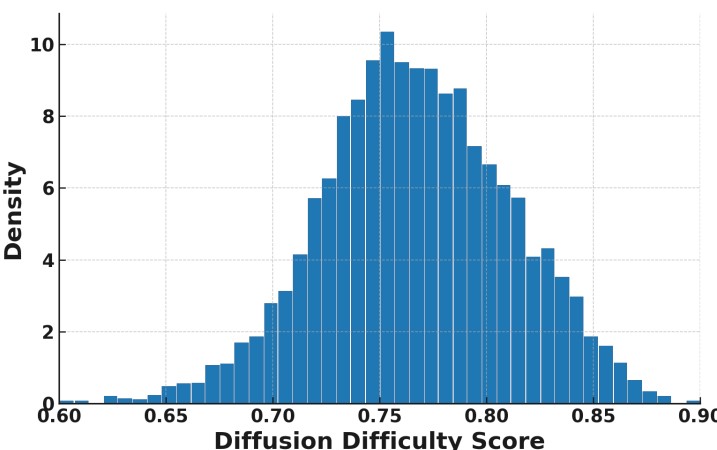

Figure 7: Distribution of diffusion difficulty score computed on LAION text–image pairs with a pre-trained SDXL model. This distribution resembles that of C2I, which supports interval sampling for selecting informative training pairs under T2I.

contexts, and the majority of samples fall in the middle range. Interval sampling remains effective for identifying informative pairs. The *Attach* phase is also easy to transfer: semantic and visual representations serve as soft supervisory signals for the selected subset.

As such, while our main experiments focus on class-to-image tasks for controlled benchmarking like SiT (Ma et al., 2024), the framework is generalizable and well suited to text-to-image generation. We expect it to deliver practical gains in data efficiency and training speed in this setting, offering a promising direction for future work.

## H  MORE DISCUSSIONS ABOUT SELECT

### H.1  DETAILED ALGORITHM FOR COMPUTING DIFFUSION DIFFICULTY SCORE

The diffusion difficulty score, used to rank samples in the *Select* phase, is defined as the mean denoising loss over uniformly sampled timesteps, computed using a frozen pretrained diffusion model (see Algorithm 2).

---

**Algorithm 2** Compute Diffusion Difficulty Score

---

**Require:** Image dataset $\mathcal{D} = \{(x_i, c_i)\}_{i=1}^{N}$; pretrained VAE encoder $E_\phi$; pretrained diffusion model $\epsilon_\theta$; timestep set $\mathcal{T}$; batch size $n$
  *// Each $x_i$ is an image; $c_i \in \{1, \dots, C\}$ is the class label. Timesteps in $\mathcal{T}$ are sampled uniformly.*
  *Models are frozen during scoring.*
 1: Initialize empty map $\mathcal{S} \leftarrow \{\}$
 2: **for** mini-batch $\{(x_i, c_i)\}_{i=1}^{n} \subset \mathcal{D}$ **do**
 3:     Encode to latent (if applicable): $z_i \leftarrow E_\phi(x_i)$
 4:     Initialize per-sample accumulator $\ell_i \leftarrow 0$
 5:     **for** $t \in \mathcal{T}$ **do**
 6:         Sample $\epsilon \sim \mathcal{N}(0, I)$
 7:         Perturb latent: $z_t \leftarrow \alpha_t z_i + \sigma_t \epsilon$
 8:         Compute loss: $\ell_i \leftarrow \ell_i + \|\epsilon - \epsilon_\theta(z_t, t, c_i)\|_2^2$
 9:     **end for**
10:     $s_i \leftarrow \ell_i / |\mathcal{T}|$  *// Mean denoising loss across timesteps*
11:     $\mathcal{S}[x_i] \leftarrow s_i$
12: **end for**
13: **Return** $\mathcal{S}$  *// Image-to-score mapping for difficulty-aware selection*

---

## H.2 Practical Insights on Interval Sampling

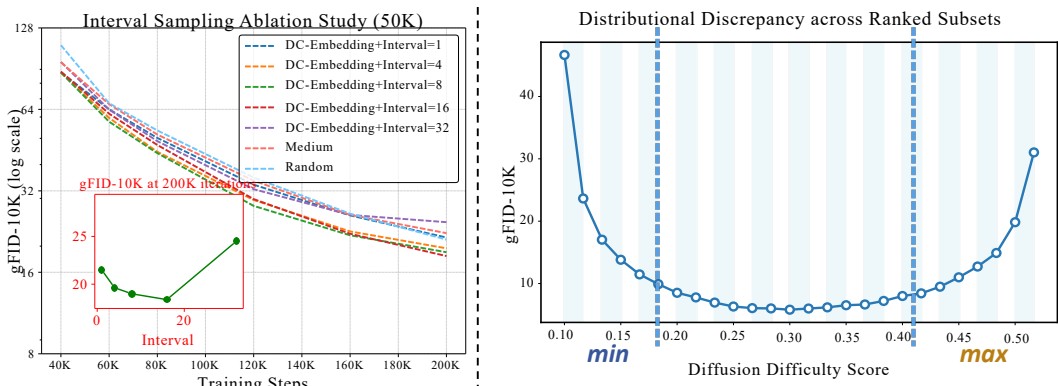

Figure 8: **Left:** gFID-10K across training steps under different interval values $k$ for a 50K data budget. Moderate intervals (e.g., $k = 16$) achieve superior performance by balancing learnability and diversity. **Right:** Distributional discrepancy (gFID-10K) between ranked training subsets and the validation set. Both extremely low and high diffusion difficulty score lead to higher FID, while mid-range segments show better alignment.

While Section 4.3 has covered a detailed ablation study on the choice of interval $k$ in *Select* phase, we provide additional insights into how diffusion difficulty score relate to distributional coverage.

The right panel in Fig. 8 presents the gFID-10K scores of subsets sampled from different portions of the difficulty-ranked dataset. We partition the training set into consecutive 10K segments ordered by the diffusion difficulty score (e.g., the first 10K samples with lowest scores as "Min", followed by 10–20K, 20–30K, and so on), and measure each segment's discrepancy from the full validation distribution using gFiD. Interestingly, we observe a clear U-shaped curve: subsets consisting of extremely low or high difficulty samples exhibit significantly worse distributional alignment, while those centered around moderate difficulty levels show substantially lower FID scores. This result aligns well with our hypothesis that very easy samples (e.g., simple textures, clean backgrounds) and extremely hard samples (e.g., ambiguous, noisy structures) both fail to reflect the global data distribution.

These observations provide an empirical justification for our interval sampling strategy. Specifically, under a 50K dataset budget with $k = 16$, each class contributes samples selected at regular intervals from its difficulty-sorted list. Given that each class typically contains around 1,200 images, this strategy naturally samples from approximately the first 800 positions in the ranked list. As a result, the selected data span both the easy and moderately difficult regions, while avoiding the extremes at both ends. This balanced coverage across the difficulty spectrum promotes better generalization and faster convergence, as evidenced by the results in Fig. 8 (Left) and discussed in Section 4.3. In this way, our strategy yields a compact yet effective dataset that enables the model to converge rapidly while maintaining strong generation quality.

# I More Discussion about Attach

## I.1 Dual Conditional Embedding

Most diffusion models condition on class identifiers represented as integer IDs or one-hot vectors, which are mapped to class embeddings trained from scratch. This ignores semantic relationships between categories, resulting in unstructured embeddings as shown in Fig. 9 (Left).In contrast, text embeddings derived from class-specific prompts (e.g., "a photo of a dog") via a pre-trained language encoder naturally encode semantic priors and cluster related classes (Fig. 9, Right). We propose a dual conditional embedding that fuses the text embedding with a learnable class embedding (i.e., a traditional class token trained from scratch), as defined in Eq. 6–7. This hybrid strategy combines semantic structure with symbolic distinctiveness, and leads to significantly improved generation quality. As shown in Fig. 6 (Right), using both branches achieves lower FID than using either one alone.

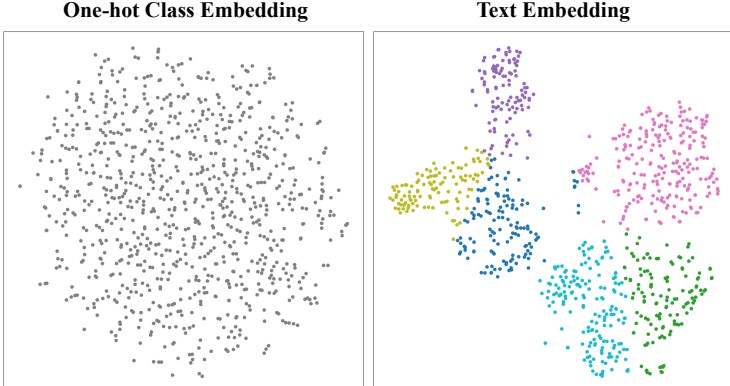

Figure 9: T-SNE visualization of class embeddings. Each point represents a class in the dataset. **Left:** One-hot class embeddings show no semantic structure. **Right:** Text embeddings naturally cluster semantically related classes. Samples from semantically related classes, such as different dog breeds, tend to form distinct clusters in feature space. Leveraging this semantic prior is highly effective for accelerating the convergence of diffusion model training.

## I.2 VISUAL INFORMATION INJECTION

Recent studies (Wu et al., 2025; Yu et al., 2025) have shown that relying solely on diffusion models to learn meaningful representations from scratch often results in suboptimal semantic features. In contrast, injecting high-quality visual priors, especially those derived from strong self-supervised encoders like DINOv2 (Oquab et al., 2023), can significantly improve both training efficiency and generation quality. In our case, we incorporate a frozen visual encoder (DINOv2) to provide external patch-level visual features during training. These external features serve as semantically rich anchors, particularly beneficial at early layers, allowing the model to focus on generation-specific details in later stages. Empirically, vi-

Table 6: Ablation of the visual encoder.

| Vision Encoder | FID↓ |
|---|---|
| N/A (baseline) | 37.07 |
| MAE-L | 9.23 |
| MoCov3-L | 8.78 |
| CLIP-L | 8.59 |
| **DINOv2-L** | **7.62** |

sual supervision improves feature alignment and accelerates convergence under limited data, as shown in Tables 1, 2, 5a, and 6. All tested encoders outperform the no-encoder baseline, indicating that our method is robust to the choice of visual encoder.

## J VISUALIZATION OF SRE$^2$L IN GENERATIVE TASKS

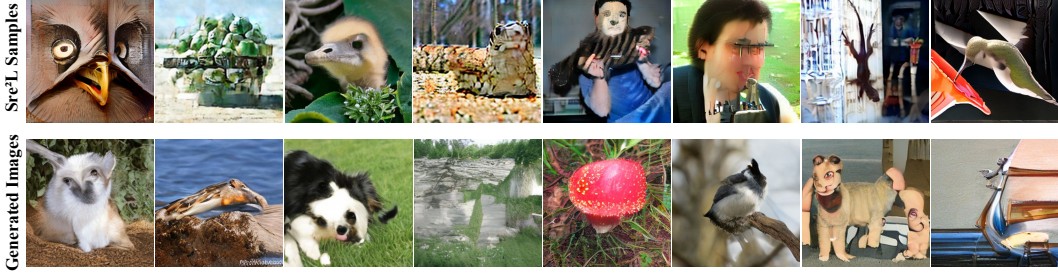

Figure 10: **Top:** images synthesized directly by SRe$^2$L, a popular dataset condensation method originally designed for discriminative tasks. **Bottom:** images generated by a diffusion model trained on the SRe$^2$L dataset. As exemplified by SRe$^2$L, such methods often struggle in generative settings—producing blurry, low-fidelity outputs that are poorly aligned with the true data distribution.

## K    IMAGENET 512×512 EXPERIMENT

As shown in Table 2, $D^2C$ consistently outperforms random sampling under a strict 10K (0.8%) data budget across both DiT-L/2 and SiT-L/2 backbones. Visual samples in Fig. 11 further confirm the high fidelity and diversity of generations at 512×512 resolution, demonstrating that $D^2C$ generalizes effectively to high-resolution settings.

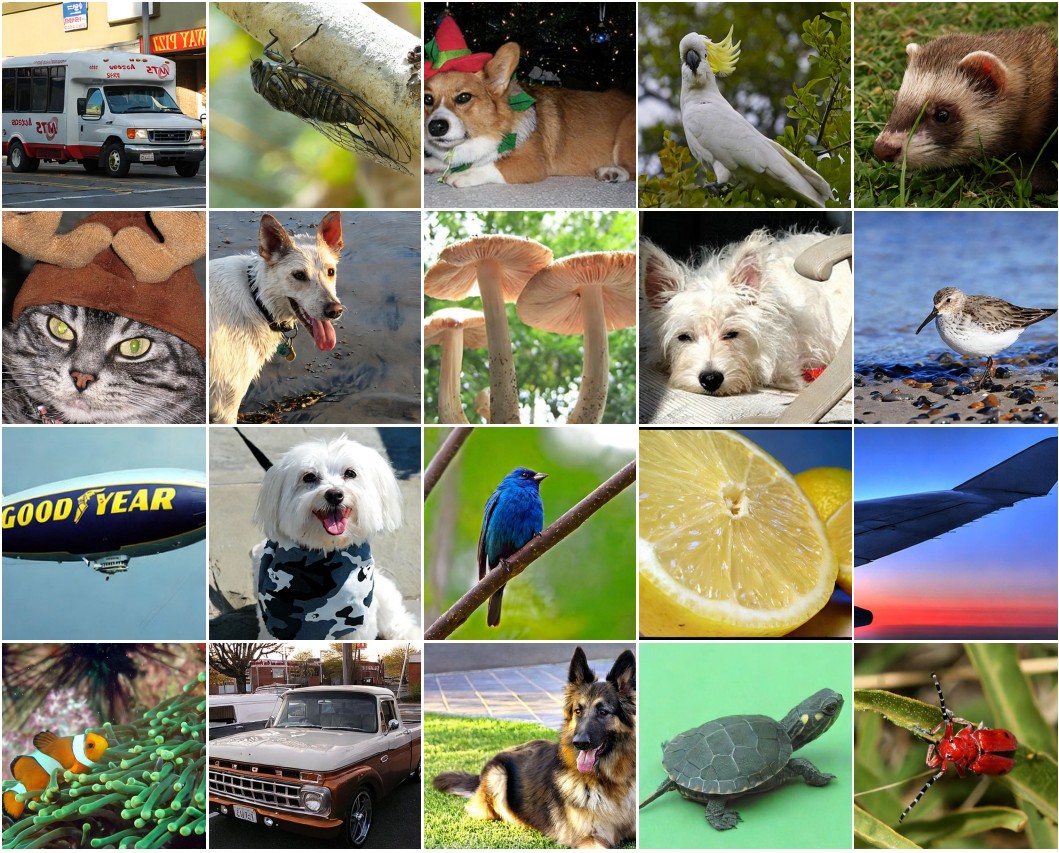

Figure 11: Generated samples on ImageNet 512×512 from SiT-L/2 trained with $D^2C$ using a 10K dataset (CFG=1.5).

## L    VISUALIZATION

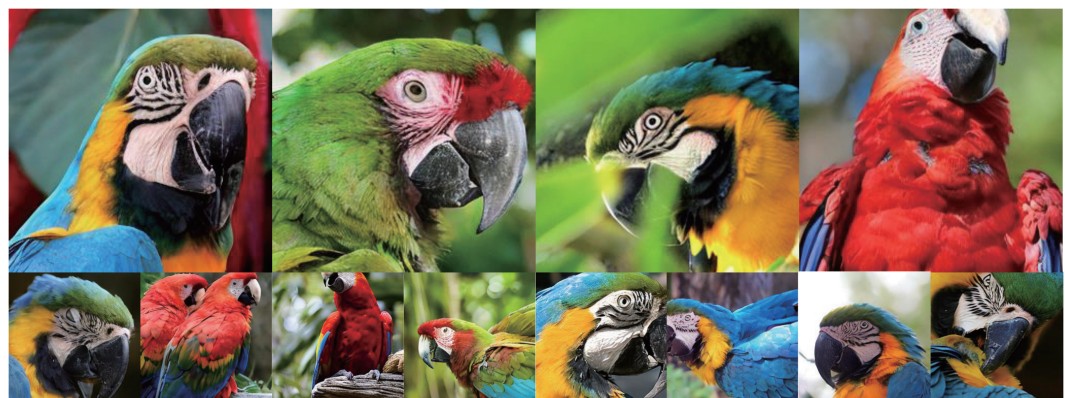

Figure 12: Generated samples of SiT-L/2 trained with $D^2C$ using a 50K dataset (CFG=1.5). Class label = "macaw"(88)

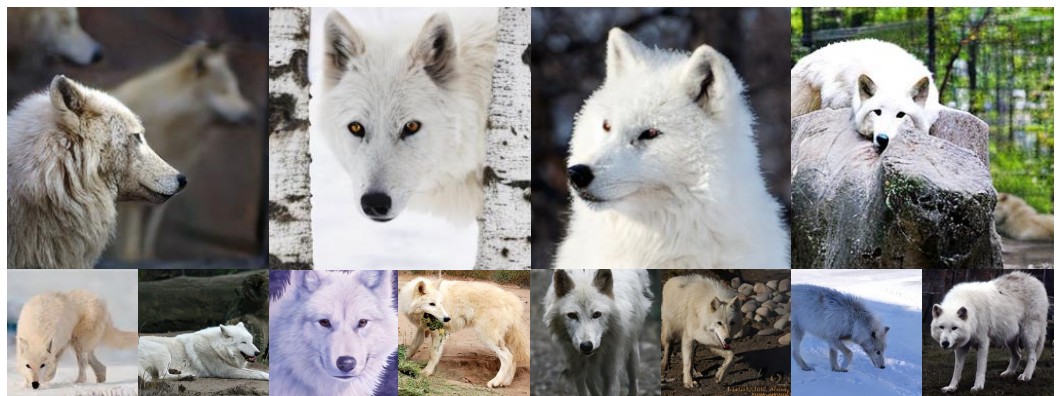

Figure 13: Generated samples of SiT-L/2 trained with $D^2C$ using a 50K dataset (CFG=1.5). Class label = "arctic wolf"(270)

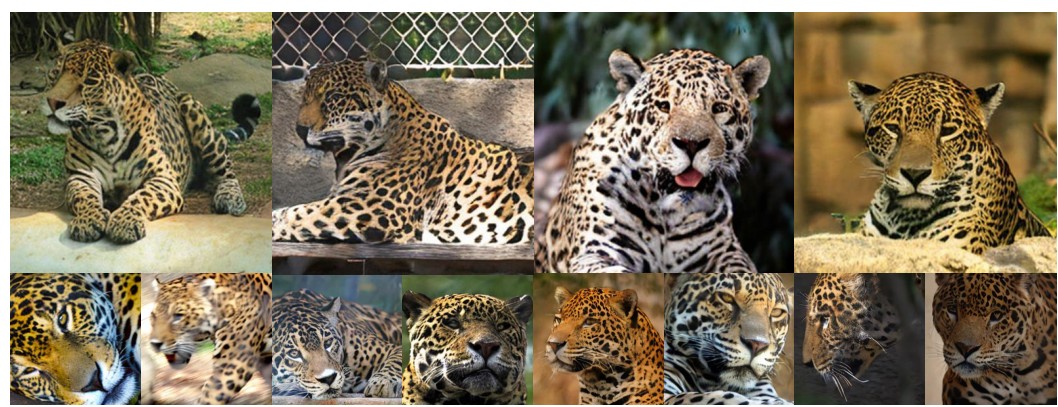

Figure 14: Generated samples of SiT-L/2 trained with $D^2C$ using a 50K dataset (CFG=1.5). Class label = "jaguar"(290)

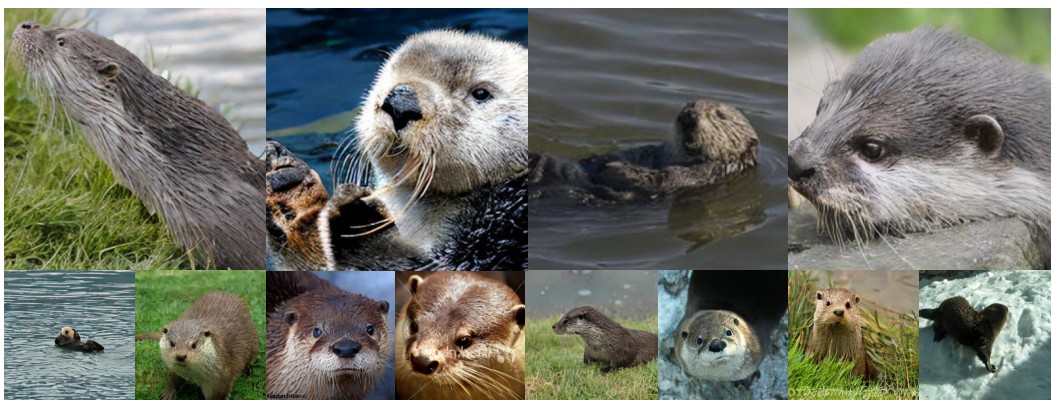

Figure 15: Generated samples of SiT-L/2 trained with $D^2C$ using a 50K dataset (CFG=1.5). Class label = "otter"(360)

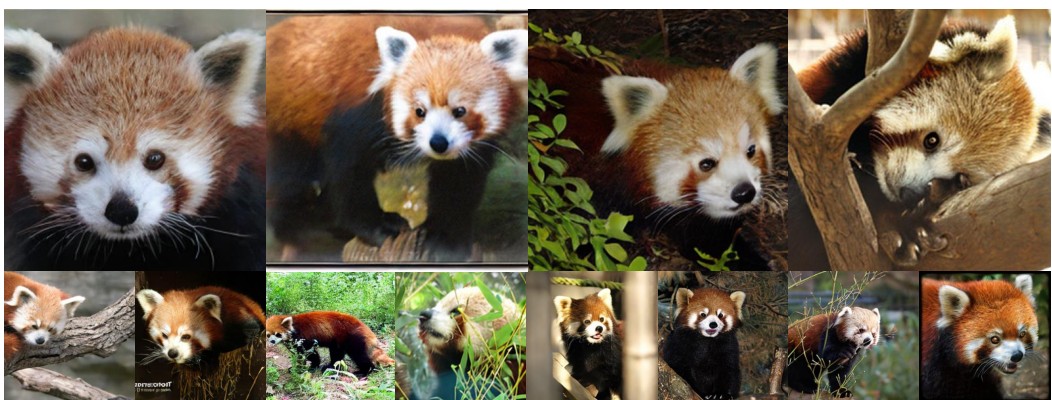

Figure 16: Generated samples of SiT-L/2 trained with $D^2C$ using a 50K dataset (CFG=1.5). Class label = "lesser panda"(387)

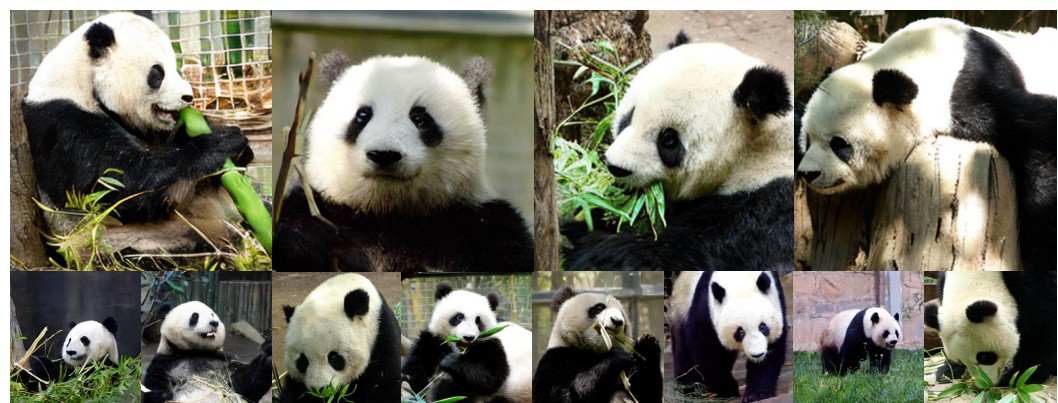

Figure 17: Generated samples of SiT-L/2 trained with $D^2C$ using a 50K dataset (CFG=1.5). Class label = "panda"(388)

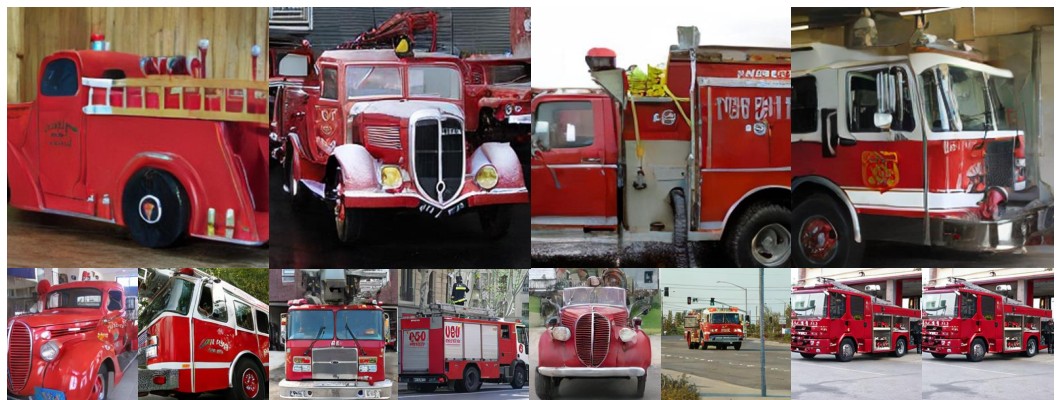

Figure 18: Generated samples of SiT-L/2 trained with $D^2C$ using a 50K dataset (CFG=1.5). Class label = "fire truck"(555)

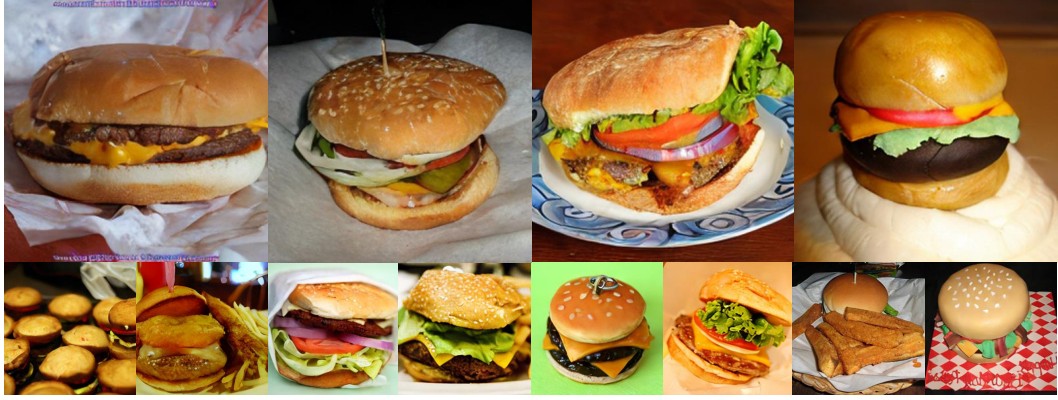

Figure 19: Generated samples of SiT-L/2 trained with $D^2C$ using a 50K dataset (CFG=1.5). Class label = "cheeseburger"(933)

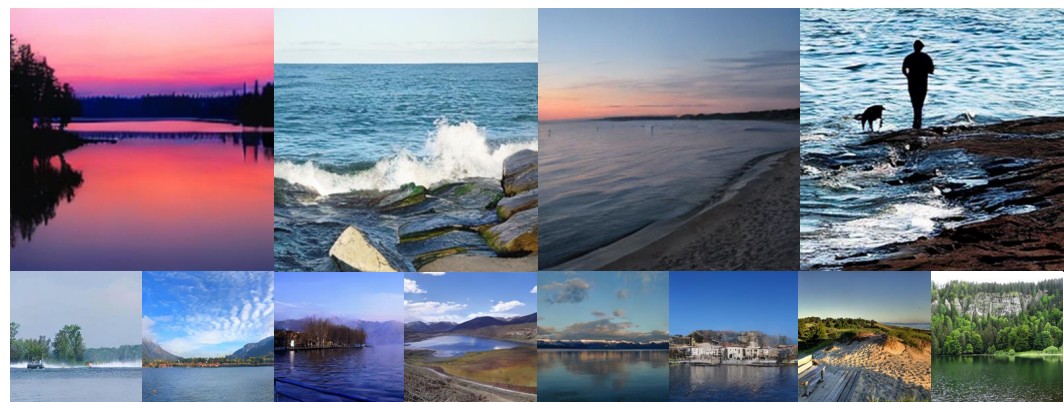

Figure 20: Generated samples of SiT-L/2 trained with $D^2C$ using a 50K dataset (CFG=1.5). Class label = "lake shore"(975)

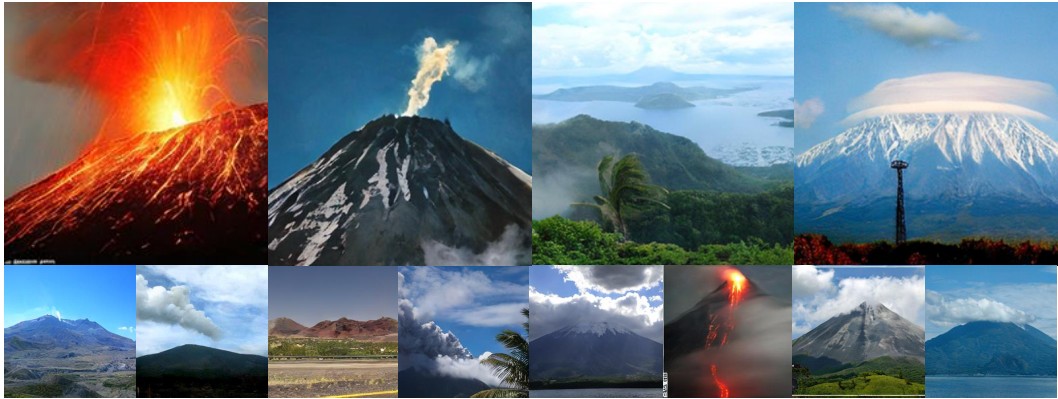

Figure 21: Generated samples of SiT-L/2 trained with $D^2C$ using a 50K dataset (CFG=1.5). Class label = "volcano"(980)

