# OpenReview forum: "Diffusion Dataset Condensation: Training Your Diffusion Model Faster with Less Data"
_ICLR.cc/2026/Conference — ICLR 2026 Conference Withdrawn Submission_

### Official Review · Reviewer_wMP5 · 2025-10-22

**Soundness:** 3
**Presentation:** 3
**Contribution:** 2
**Rating:** 4
**Confidence:** 4

**Summary:**

The paper proposes a dataset condensation method for diffusion model training. The method consists of two phases: select and attach. In the select phase, the paper ranks the difficulties of the given data using the score from the pretrained model, then selects at a fixed interval. In the attach phase, the paper attaches additional information such as text encoding and visual information injection. With the condensed small data, the paper trains a model with both a denoising loss and a semantic alignment loss.

**Strengths:**

With only a small amount of data, the method achieves performance comparable to REPA and vanilla models while showing much faster convergence. In addition, the approach of constructing the dataset through interval sampling based on diffusion model difficulty is intuitive and easy to understand. This research topic has the potential to significantly improve the accessibility of diffusion studies, which have traditionally required substantial computational resources.

**Weaknesses:**

- It seems that the performance improvement of this dataset may come not from the selection process itself, but rather from attaching additional information. As shown in Table 5, gFID score is already comparable even without the selection step. Adding additional descriptions appears to help the model learn faster and achieve quicker convergence, making this work more similar to studies that investigate enriching data with additional information, such as long captions.
- Comparisons with other dataset cleaning, distillation, or augmentation methods are needed. Although many of these works have been explored primarily in the GAN domain (e.g., Instance Selection for GANs), it would be better to demonstrate that this selection approach is more effective than using datasets selected by those existing methods. Since the method already relies on pretrained models, it can be compared with works that use them to build distilled datasets (e.g., D4M: Dataset Distillation via Disentangled Diffusion Model, CVPR 2024).
- While the paper argues the proposed method is data condensation, it seems there is no image optimization unlike other condensation methods that synthesize original data using gradient-based optimization.

**Questions:**

- Would it not be sufficient to randomly select images and then attach additional information to them?
- Since the images remain raw data, and only additional information is attached, so the work may be similar with data cleaning or pruning methods.
- Could comparisons with other dataset augmentation or distillation methods be added?

---

### Official Review · Reviewer_UBi3 · 2025-10-28

**Soundness:** 3
**Presentation:** 3
**Contribution:** 3
**Rating:** 6
**Confidence:** 3

**Summary:**

This paper addresses the significant computational and data cost associated with training high-quality diffusion models. The authors introduce "diffusion dataset condensation" as a new problem setting and propose a novel two-stage framework called Diffusion Dataset Condensation to tackle it.

**Strengths:**

1.⁠ ⁠The paper's primary strength is identifying and formalizing the problem of dataset condensation specifically for diffusion models. It points out that prior methods for discriminative tasks fail in the generative setting (as shown in Table 4), necessitating a new approach.

2.⁠ ⁠⁠The ability to achieve a 4.3 FID on ImageNet 256*256 using only 10K samples (0.8% data) and 40K training steps is an excellent result. The massive speedups in both training steps (100x-233x) and wall-clock time (21x-101x) are highly significant.

3.⁠ ⁠⁠The D_2C framework is well-designed. The two-stage Select and Attach process is logical. The Select phase's use of a difficulty score to balance learnability and diversity is more sophisticated than naive pruning. The Attach phase's fusion of semantic and visual priors is a proven way to boost data efficiency. The method is also modular; as shown in the ablations, each part contributes value.

4.⁠ ⁠⁠Another key strength worth mentioning is that the paper is well written, all ablations and dimensions are experimented in detail.

**Weaknesses:**

1. The Attach phase, while effective, has limited novelty. The visual information injection and the L_proj loss are explicitly adopted from REPA. The DC-Embedding's fusion of learnable and pre-trained text embeddings is a well-known and effective technique. This places the bulk of the method's novelty on the Select phase and the combination of these parts for this new task.

2.⁠ ⁠The fixed interval sampling seems trivial. Why not try other ablations over the sampling that respect the true distribution of the data?

**Questions:**

1.⁠ ⁠The comparison in Table 5b shows a significant difference between the result achieved using a strong pretrained scorer (4.3 gFID) versus a weaker scorer trained from scratch (4.9 gFID). Does this suggest that the performance ceiling of D2C is fundamentally linked to the quality of the initial model, potentially transferring the heavy computational burden from the diffusion model training phase to the scorer training phase?

2.⁠ ⁠Did you try other ablations over the sampling techniques?

---

### Official Review · Reviewer_yCdG · 2025-10-31

**Soundness:** 1
**Presentation:** 2
**Contribution:** 1
**Rating:** 2
**Confidence:** 5

**Summary:**

The paper proposes a two-stage data condensation method for diffusion models. The first step is selection, based on the difficulty (negative conditional probability). The second step is attachment, based on the semantic information (Dual Conditional Embeddings and Visual Features). For experiments, D2C is compared with traditional data selection methods like Herding, Random sampling, and K-Center. Also, a distillation method SRe2L is compared with D2C. It shows acceleration on diffusion models empirically.

**Strengths:**

1. It studies the problem of data condensation in the diffusion models.
2. The paper is easy to follow.

**Weaknesses:**

1. Positioning and Novelty: The paper's claim regarding the novelty of studying data efficiency in diffusion models appears to be overstated. For instance, recent work (e.g., [arXiv: 2409.19128]) also investigates this specific problem. The authors are strongly encouraged to thoroughly discuss this and other related prior art, situating their own contributions more accurately within the existing literature to avoid overclaiming.

2. Methodological Validation: The methodological contributions require further justification and validation through more rigorous ablation studies:
- Selection Mechanism: The paper employs conditional probability for data selection but does not benchmark this choice against other established metrics. To justify this design, an ablation study is needed, comparing the proposed method to alternatives such as Gradient Norm or Loss-based scores.
- Attachment Mechanism: The effectiveness of the attachment step seems highly dependent on the external feature/embedding extraction model. It is currently unclear whether the performance gains stem from the proposed attachment mechanism itself or simply from the rich semantics provided by the external model. To isolate the mechanism's specific contribution, the authors should conduct an experiment that removes or replaces this external model (e.g., using simpler, non-external embeddings) and compares the performance.

3. Empirical Evaluation and Baselines: The empirical validation for the D2C method is not sufficient to be convincing, as the experiments are missing comparisons against several key state-of-the-art (SOTA) methods.

- Selection Baselines: The comparison is limited to methods like Herding, Random, and K-center. The evaluation would be substantially stronger if it included more recent and relevant data selection techniques, such as CCS [1] and TDDS [2].
- Distillation Baselines: The distillation experiments rely solely on SRe2L, which is insufficient. This comparison omits numerous modern methods like DC [4], DM [5], RDED [6], and NCFM [7].



[1] Zheng H, Liu R, Lai F, et al. Coverage-centric coreset selection for high pruning rates[J]. arXiv preprint arXiv:2210.15809, 2022.
[2] Zhang X, Du J, Li Y, et al. Spanning training progress: Temporal dual-depth scoring (tdds) for enhanced dataset pruning[C]//Proceedings of the IEEE/CVF Conference on Computer Vision and Pattern Recognition. 2024: 26223-26232.
[3] Zhao B, Mopuri K R, Bilen H. Dataset condensation with gradient matching[J]. arXiv preprint arXiv:2006.05929, 2020.
[4] Zhao B, Bilen H. Dataset condensation with distribution matching[C]//Proceedings of the IEEE/CVF Winter Conference on Applications of Computer Vision. 2023: 6514-6523.
[6] Sun P, Shi B, Yu D, et al. On the diversity and realism of distilled dataset: An efficient dataset distillation paradigm[C]//Proceedings of the IEEE/CVF Conference on Computer Vision and Pattern Recognition. 2024: 9390-9399.
[7] Wang S, Yang Y, Liu Z, et al. Dataset distillation with neural characteristic function: A minmax perspective[C]//Proceedings of the Computer Vision and Pattern Recognition Conference. 2025: 25570-25580.

**Questions:**

Please see Weaknesses for details.

I would consider update my scores if the authors conduct extensive results and add related discussion and references properly.

---

### Official Review · Reviewer_mRdu · 2025-10-31

**Soundness:** 2
**Presentation:** 2
**Contribution:** 2
**Rating:** 2
**Confidence:** 4

**Summary:**

This paper introduces Diffusion Dataset Condensation (D2C), a framework of dataset distillation for diffusion models. The key idea is to construct a small yet information-rich synthetic sub-dataset that enables high-quality diffusion model training with only a fraction of the original data. D2C contains two phases: Select phase and Attach phase. The experiments show that only using 0.8% data the diffusion model can be trained from scratch and achieve a good performance.

**Strengths:**

1.  This is the first paper to formally study dataset condensation for diffusion models, whereas prior works (e.g., SRe2L, MTT, CAFE) targeted discriminative tasks like classification.

  2.  D2C achieves up to 233× faster training using only 0.8% of ImageNet data, while maintaining competitive FID (e.g., 4.3 at 40k steps).

**Weaknesses:**

1.  Although the diffusion model itself is trained from scratch, the method relies heavily on pretrained T5 and DINOv2 encoders in the Attach phase, and a pretrained DiT for scoring. It is uncertain that the performance gain is from the data or the pretrained models.

  2. The paper is mainly empirical and there is no theoretical guarantee or formal analysis.

  3. The performance of the CS divergence is sensitive to the choice of kernel and bandwidth, but the paper does not provide a robustness analysis to assess this sensitivity.

  4. The effectiveness of the D2C is unclear. For example, the random selection reported in Table 1 can achieved 4.19  gFID-50K, while the D2C only achieves 4.13. I do not see any performance gain from the D2C.

  5. Another point concerns the ablation study. Table 5(a) shows that the Attach phase contributes the most to the overall performance improvement. However, according to the random selection results in Table 1, even the random selection  achieves strong performance. This raises a question about the marginal contribution of the Select phase versus the Attach phase, and whether the observed gains primarily stem from the added semantic and visual augmentations rather than from the data selection itself.

**Questions:**

1. The table 1 mentioned that the experiments are reporting various dataset condensation methods. However,  K-Center, Herding are coreset selection methods not dataset condensation methods. The authors should include the baselines of dataset condensation such as MTT.

2. The performance advantage of the SRe2L baseline primarily arises from its use of soft labels in training discriminative models. However, diffusion models rely on discrete class labels during training, meaning that SRe2L’s mechanism cannot be directly applied in this setting. Without soft labels, SRe2L would likely perform poorly, making it an inappropriate or unfair baseline for comparison in the diffusion model context.

---

### Official Review · Reviewer_B6cT · 2025-11-06

**Soundness:** 3
**Presentation:** 3
**Contribution:** 2
**Rating:** 4
**Confidence:** 4

**Summary:**

This paper introduces Diffusion Dataset Condensation (D$^2$C), a novel two-stage framework aimed at significantly reducing the data and computational requirements for training diffusion models. The method consists of a Select phase, which uses a diffusion difficulty score and interval sampling to curate a compact yet diverse subset of training images, and an Attach phase, which enriches the selected samples with semantic (text) and visual (patch-level) embeddings. Extensive experiments on ImageNet-1K at various resolutions and model architectures demonstrate that D$^2$C achieves substantial acceleration and maintains competitive FID scores with data compression ratios as low as 0.8%.

**Strengths:**

(1) The paper is well-written and easy to follow. The motivation is clearly articulated, and the experimental setup is presented transparently.

(2) The proposed D$^2$C framework is technically sound and intuitive. The decomposition into "Select" and "Attach" stages seems logical: The "Select" stage's use of a "diffusion difficulty score" is for identifying informative samples and the subsequent interval sampling to ensure diversity is a simple yet effective addition. Furthermore, in the "Attach" stage, the authors effectively increase the information density of each sample, guiding the model to learn more efficiently by "attaching" richer conditioning signals (both semantic and visual).

(3) Given the massive computational costs associated with training SOTA diffusion models, an efficient method in this domain would have a substantial practical impact on inspiring similar efficiencies in other generative frameworks like flow-based models.

**Weaknesses:**

(1) The "Attach" stage's augmentation with semantic and visual representations is innovative but lacks theoretical justification. This could be improved by deriving bounds or analyses showing how attachments reduce information loss in condensed datasets.

(2) The term "Dataset Condensation" (DC) traditionally implies creating a small set of synthesized samples (e.g., through gradient matching) that are optimized to train a model. This paper, however, selects existing samples from the original dataset. While the "Attach" stage enriches these samples, it doesn't synthesize the image pixels themselves. Therefore, the method is more accurately described as a form of "Informed Data Selection and Enrichment" rather than "Condensation" in the classical sense. This naming could be misleading in DC literature. A clarification of this distinction is necessary.

(3) Experimental comparisons seem strong but lack diversity in datasets beyond ImageNet-1K; testing on more varied domains like CelebA (for faces) or LSUN (for scenes) would better demonstrate generalization, as ImageNet's class-balanced structure might bias results toward structured data.

**Questions:**

(1) Could authors clarify on choice of the term "Dataset Condensation" given that D$^2$C selects, rather than synthesizes, data samples? A brief discussion in the paper clarifying this and positioning your work with respect to classic DC would be very helpful.

(2) It should be noted that the proposed approach is designed for and limited to the ImageNet dataset. Consequently, applying D$^2$C to Stable Diffusion models trained on LAION data for text-to-image generation poses significant challenges, primarily due to the partial domain shift.

(3) What are the potential advantages and emergent capabilities arising from scaling up image resolution to 1024, extending the model architecture to an XL variant, or even applying video datasets for training Video diffusion models?

(4) The paper mainly emphasizes the acceleration of the training phase. However, it does not analyze the computational cost of the D$^2$C pipeline itself. This pre-processing includes: (i) a full inference pass over the entire original dataset with a large diffusion model, (ii) generating rich captions for the selected subset, and (iii) encoding the selected images.
For a fair comparison of overall efficiency, this pre-processing cost must be quantified and compared against the training time saved. When training on smaller datasets, the D$^2$C 's overhead could be substantial.

---

### Note · Authors · 2025-11-14

I have read and agree with the venue's withdrawal policy on behalf of myself and my co-authors.